# Soil Quality Changes within a *(Nothofagus obliqua)* Forest Under Silvopastoral Management in the Andes Mountain Range, South Central Chile

**Juan Ortiz** [1,2]**, Francis Dube** [1,]*** , Pablo Neira** [1]**, Marcelo Panichini** [3]**, Neal B. Stolpe** [4]**,
Erick Zagal** [4] **and Pedro A. Martínez-Hernández** [5]

[1] Department of Silviculture, Faculty of Forest Sciences, University of Concepción, Victoria 631, Casilla 160-C, 4030000 Concepción, Bio Bio Region, Chile; jortizc@udec.cl (J.O.); paneirav@gmail.com (P.N.)

[2] Doctoral Program in Forest Sciences, Faculty of Forest Sciences, University of Concepción, Victoria 631, Casilla 160-C, 4030000 Concepción, Bio Bio Region, Chile

[3] National Agricultural Research Institute (INIA) Quilamapu, Vicente Méndez 515, 3812120 Chillan, Ñuble Region, Chile; marcelo.panichini@inia.cl

[4] Soils and Natural Resources Department, Faculty of Agronomy, University of Concepción, Vicente Méndez 595, Casilla 537, 3812120 Chillan, Ñuble Region, Chile; nstolpe@udec.cl (N.B.S.); ezagal@udec.cl (E.Z.)

[5] Department of Animal Husbandry, Chapingo Autonomous University, Km. 38.5, 56230 Mexico-Texcoco Road, Mexico; pedroarturo@correo.chapingo.mx

\* Correspondence: fdube@udec.cl; Tel.: +56-41-220-4982

**Abstract:** In Chile, 49.1% of the national territory is affected by soil degradation (including erosion and loss of soil organic matter), whereby of the 51.7 Mha that have been historically associated with agricultural-livestock and forestry activities, only 35.5 Mha are being used at the present. Consequently, soil degradation has resulted in the release of about 11.8 Gg $yr^{-1}$ of carbon (C) equivalent ($CO_{2eq}$) to the atmosphere. Silvopastoral systems (SPS), however, can increase soil organic C (SOC) through sequestration (C→SOC), improve ecosystem services, and have been internationally recommended for sustainable land use. Therefore, it was proposed to determine the effects of SPS on soils, over five years, in degraded sites that were located in the Ranchillo Alto (SPS-RA) (37°04′52″ S, 71°39′14″ W), Ñuble region. The sites were rated according to previous canopy disturbance levels (+) as follows: open ($O_p$)+++, semi open ($SO_p$)++, and semi closed (SC)+. The analysis was performed on different physical and chemical soil properties (0–5 and 5–20 cm depths), that were expressed as soil indicators ($S_{IND}$) for chemical and physical properties, which were used to calculate a soil quality (SQ) index (SQI). The results indicated overall SQI values of 37.6 (SC) > 29.8 ($O_p$) > 28.8 ($SO_p$), but there were no significant variations ($p < 0.05$) in physical SQ, whereas chemical SQ varied in all conditions, mostly at 0–5 cm in $O_p$ and $SO_p$. Increases of SOC were also observed (2015–2018 period) of 22.5, 14.5, and 4.8 Mg $ha^{-1}$ for $SO_p$, $O_p$, and SC, respectively, showing that SPS promote the reclamation of Ranchillo Alto soils.

**Keywords:** agroforestry systems; sustainable land management; C sequestration; Andisols

## 1. Introduction

At present, of the 10–12 Pg $yr^{-1}$ of the world emissions of carbon equivalent ($CO_{2eq}$) (greenhouse gas emissions expressed as $CO_2$) [1,2], deforestation is the second largest anthropogenic source of $CO_2$ emissions [3]. Additionally, of the 4033 Mha that comprise the world forest areas, around 3.24% has been subjected to logging, which has resulted in the release of approximately 20% of the global soil

organic C (SOC) depletion, mainly due to the expansion of agricultural frontiers [4]. Specifically, South America is the second leading region of the world for $CO_{2eq}$ emissions from agricultural-forestry and livestock practices (40%) [5], resulting in losses of approximately 8.7% of the forest areas [4].

As a part of the international efforts addressing climatic change, silvopastoral systems (SPS), which are defined as predetermined associations of woody and herbaceous species and livestock and are a subtype of an agroforestry system (AFS), have remarkable carbon sequestration (C→SOC) potential, and are able to store around 1.8–6.1 Mg SOC $yr^{-1}$ [6–8], which is critically relevant to natural pedologic processes, management practices, and environmental functions. In this regard, Karlen et al. [9], introduced the concept soil quality defined as, "Soil capacity under a determined management or ecosystem fringe to sustain biological productivity, preserve environmental functions, promoting plant and animal development and consequently human health". Soil quality (SQ) is measured through SQ indicators ($S_{IND}$), which directly or indirectly reflect soil functionality at different timescales [10,11]. According to [12–14], the selection criteria for soil indicators ($S_{IND}$) should include the following: (i) a correlation with ecosystem processes (e.g., C→SOC); (ii) integration with chemical physical and biological properties, (iii) easily measured, replicated, and verified; (iv) a sensitivity to seasonal or atmospheric variations and realistic management practices; (v) compatibility with previous data; and (vi) usefulness for different professionals.

Regarding the specific case of Chile, decades of overutilization of natural resources has resulted in 49.1% of the national territory being affected by soil degradation [15], which has caused yearly emissions of approximately 11.8 Gg $CO_{2eq}$, of which approximately 38% comes from agricultural soils [16]. Moreover, of the 51 Mha that have been historically used in agricultural-forestry and livestock production, only 35.5 Mha presently remain active [17], mostly because of a massive loss of forest biomass that has had a critical effect on optimal soil functionality (e.g., erosion) [18]. On that basis, different government and institutional initiatives have been implemented (e.g., National Forest Program), in order to mitigate soil degradation. However, SPS have achieved only a limited presence in the regulatory framework, a fact that has been documented by scientific researchers [19]. In Coyhaique, of the Chilean Patagonia, [20] an investigation compared SPS to an introduced plantation (both comprised of *Pinus ponderosa*), with results that showed C stocks of 224 Mg C $ha^{-1}$, 199 Mg C $ha^{-1}$, and a net C accumulation of 1.8 Mg $yr^{-1}$ (800 trees) and 2.5 Mg $yr^{-1}$ (400 trees) for the SPS and plantation, respectively. The authors concluded that trees in SPS use the site resources more efficiently (up to 30%).

Therefore, it was proposed to study the SPS within a native Roble (*Nothofagus obliqua*) forest in the Region of Ñuble (SPS-RA) having distinct levels of degradation, with the objective to determine the effect of SPS over the physical and chemical aspects of SQ, after five years of establishment. It was hypothesized that the SQ index (SQI) in the SPS-RA, would tend to increase at depths of 0–5 cm as a result of improved silvopastoral management, with annual net accumulation of SOC, regardless of the initial site condition, resulting in values above the minimal range reported in the literature for SPS (1.8 Mg SOC).

## 2. Materials and Methods

### 2.1. Description of the Study Site

The silvopastoral systems (SPS) are located in the Ranchillo Alto area which is a state-owned property in the Ñuble region (37°04′52″ S, 71°39′14″ W; 1200–2000 m.a.s.l) covering an area of about 635 ha and 120 km east of the City of Concepcion [19,21].

The silvopastoral systems located in the Ranchillo Alto area (SPS-RA) comprise 24 ha and were established mainly to recover the ecosystem value of the native forest along with the promotion among the community of sustainably oriented, rural economic practices. The woody element in the SPS-RA is Roble (*Nothofagus obliqua*), while the herbaceous component includes oats (*Avena sativa*), vetch (*Fabaceae purpurea*), clover (*Trifolium incarnatum*, *T. subterraneum* y *T. vesiculosum*), *Lolium multiflorum westerwoldicum*, *Phalaris acuatica*, *Lolium perenne*, *Festuca arundinacea*, *Dactylis glomerate*), and the re-sprouting of Radal (*Lomatia hirsuta*), and Quila (*Chusquea quila*). According to USDA, 2014 [22], the

soils are Andisols, "Santa Barbara" series (medial, amorphic, mesic Typic Haploxerands), and locally known as "trumaos".

Andisols in Chile are of major importance for agricultural production, corresponding to approximately 60% of national arable land (2.5 Mha) [23]. Moreover, Andisols constitute about 30–70% of the total surface in the Andean mountain range, which have critical relevance in terms of water cycling (e.g., preventing potential flooding downstream) [24,25]. However, previous conditions in the SPS-RA include over grazing and browsing and excessive logging, generating degradation processes evidenced by discontinuous soil cover, topsoil removal, formation of gullies, and massive losses of soil organic matter (SOM) [26].

To determine the SQI in the SPS-RA, the respective SQ indexes ($S_{IND}$) were examined and grouped as follows: (i) chemical parameters such as pH, %SOC, total N, $NH_4^+$, $NO_3^-$, P, $K^+$, $Ca^{2+}$, $Mg^{2+}$, $Na^+$, S, exchangeable Al ($Al_{EXCH}$), % of Al saturation (%$Al_{SAT}$) that are linked to soil fertility, and therefore to C→SOC and (ii) physical parameters such as particle density (PD), bulk density (BD), total porosity % ($P_{OR}$), % of water stable aggregates (WSA), infiltration velocity ($I_{NF}V$), water holding capacity (WHC) and penetration resistance ($P_{EN}R$) which couple soil particle arrangement and environmental services (e.g., water cycling).

## 2.2. Soil Sampling and Analysis

A total of thirty-six soil samples, randomly chosen, were collected in January 2019 from the depths of 0–5 and 5–20 cm for the three considered conditions (open ($O_p$), semi open ($SO_p$), and semi closed (SC)) (Table 1). Each sample (analyzed in triplicate), was air-dried, mixed, ground, and passed through a 2 mm sieve for determination of the respective $S_{IND}$. The work was carried out at the Agricultural Research Institute (INIA, Quilamapu) to determine most of the properties (except for SOC% and N%).

The chemical $S_{IND}$ of SOC and total N were analyzed at the Soil and Natural Resources Laboratory (Faculty of Agronomy, University of Concepción), according to Wright and Bailey (2001) [27]. The temporal variation of SOC and N was determined by comparison of the current to previous data from the sites. The remaining indicators of pH$_{(water)}$, %SOM, [$NH_4^+$, $NO_3^-$, P, $K^+$, $Ca^{2+}$, $Mg^{2+}$, effective cation-exchange capacity (ECEC), $Na^+$, S, $Al_{EXCH}$] concentrations, and %$Al_{SAT}$ were conducted using the methods proposed by Sadzawka et al. (2006) [28]. Regarding the physical $S_{IND}$, %WSA (water stable aggregates) was measured according to Kemper and Rosenau (1986) [29], where soil samples were placed in a 0.250 mm sieve and immersed within an aluminum chamber containing distilled water during 3 min, with a cycling of 1.3 cm (35 rep min$^{-1}$). The dispersed soil was placed in containers and dried at 105 °C, while the remaining soil was re-immersed into an aluminum chamber containing sodium hexametaphosphate (2 g L$^{-1}$) during 15 min with a cycling of 1.3 cm (35 rep min$^{-1}$). Once dried, samples from both procedures were weighed in order to determine each proportion within the total sample. The WHC (water holding capacity) was determined according to Zagal et al. (2003) [30]. A sample with a 1:2 soil water ratio was placed into a plastic cone sealed with adhesive tape at the bottom for about 12 h, after which the tape was carefully perforated to allow the water to drain which was collected into a plastic bottle, and then the subsequent liquid volume was measured. The $P_{EN}R$ determination was carried out by using a penetrometer model Soil Compaction Tester Dickey-John (Auburn, IL, USA); and determinations were made by following a transect over the 60 plots for each site condition in order to achieve a reliable representativity. Field measurements of unsaturated hydraulic conductivity (K) were performed using an infiltrometer model Mini Disk Infiltrometer S (Pullman, WA, USA). The methodology proposed by Zhang (1997) [31], was used to determine K (cm day$^{-1}$), based on the cumulative infiltration measurements. Bulk density (BD), was measured in soil that was sampled using cylindrical soil cores (211 cm$^3$), which were subsequently dried at 105 °C until reaching a constant weight, [32]. The soil particle density (PD) was evaluated through the pycnometer method [33], and net pore space (P)% was calculated from BD and particle density (PD) values, using the following equation:

$$P = [PD − BD/PD] \times 100 \tag{1}$$

**Table 1.** General information for each tree cover condition in the silvopastoral systems located in the Ranchillo Alto area (SPS-RA).

| Cond | Location | Total Area (ha) | N° P and Area (ha) | Tree Density (N° ha$^{-1}$) | Forest Species | Tree Cover Description | Previous Degradation | Soil Sampling |
|---|---|---|---|---|---|---|---|---|
| O$_P$ | 37°14′51″ S, 72°26′30″ W 1250 m.s.n.m | 4 | 3 × 1.33 | 60 | Roble (*Nothofagus obliqua*) | Ground with 85–95% of external light (average area) | +++ | 2 depths (0–5 and 5–20 cm) × 6 sampling points |
| SO$_P$ | 37°14′50″ S, 72°26′30″ W 1250 m.s.n.m | 4 | 3 × 1.33 | 134 | Roble (*Nothofagus obliqua*) | Soil with 65–75% of external light (average area | ++ | 2 depths (0–5 and 5–20 cm) × 6 sampling points |
| SC | 37°14′49″ S, 72°26′30″ W 1250 m.s.n.m | 4 | 3 × 1.33 | 258 | Roble (*Nothofagus obliqua*) | Soil with/ 45–55% of external light (average area) | + | 2 depths (0–5 and 5–20 cm) × 6 sampling points |

* Cond, condition; N° P, number of plots. A, area

### 2.3. Soil Quality Assessment

Soil quality estimation was performed by the selection of different $S_{IND}$, based on their relevance to the properties of the SPS-RA (e.g., soil fertility reclamation, C→SOC), and the previously stated selection criteria. Once a specific $S_{IND}$ was analytically characterized (in the field or laboratory), a numerical point value (score) was, then, assigned to it, based on qualitative ranges (low, medium, and high) reported in the literature (e.g., Amacher et al. (2007) and Vidal (2007) [34,35]). Those ranges were related to different soil functionality levels, from critical to optimal, in which an individual $S_{IND}$ can influence the overall status of soil quality SQI. However, in this study, the overall SQI was subdivided into chemical ($SQI_{CHEMICAL}$) and physical ($SQI_{PHYSICAL}$) and most of the ranges (low, medium, and high) for any single $S_{IND}$ were taken from [34], although in some $S_{IND}$, the ranges were adjusted with the aim of improving their local representativity, based on the national scientific literature and the unique properties of Andisols. Additionally, another $S_{IND}$ was included from the original proposal that was conducted by [34], in order to calculate the proposed global SQI (see Appendix A) for the specific purposes of this study. Accordingly, the chemical SQI was calculated as follows:

$$SQI_{CHEMICAL} = \Sigma \, [pH + \%SOC + CEC + N + NH_4^+ + NO_3 \\ + C{:}N + P + K + Ca^{2+} + Mg^{2+} + Na + S + Al_{EXCH} + Al_{SAT}] \tag{2}$$

where $SQI_{CHEMICAL}$ is the chemical soil quality index, %SOC is the percentage of SOC, ECEC is the effective cation-exchange capacity, N is total N, $NH_4^+$ is available ammonium, $NO_3$ is available nitrate, C/N is the C:N ratio, P is available phosphorus, $K^+$ is potassium content, $Ca^{2+}$ is calcium content, $Mg^{2+}$ is magnesium content, Na is sodium content, $Al_{EXCH}$ is exchangeable aluminum, and $Al_{SAT}$ is aluminum saturation (%).

Subsequently, physical SQI by $S_{IND}$ was calculated as follows:

$$SQI_{PHYSICAL} = \Sigma \, [I_{NF}V + \%WSA + WHC + P_{EN}R + BD + PD + P_{OR}] \tag{3}$$

where $SQI_{PHYSICAL}$ is the physical soil quality index, $I_{NF}V$, is infiltration, %WSA is water stable aggregates %, WHC is water holding capacity, $P_{EN}R$ is penetration resistance, BD is bulk density, PD is particle density, and $P_{OR}$ is total porosity.

Therefore, a global SQI was estimated using the means of both SQI types as follows:

$$SQI_{GLOBAL} = \Sigma \, [SQI_{CHEMICAL} + SQI_{PHYSICAL}]/2 \tag{4}$$

Finally, the % valuation for any site condition was calculated as:

$$\% \, SQ = [\text{number of } S_{IND} \text{ at critical level/number of } S_{IND} \text{ estimated}] \times 100 \tag{5}$$

\* A list of abbreviations is provided in Appendix B (Table A3).

### 2.4. Statistical Analyses

The data were input, and calculations made for each $S_{IND}$, and the conversion to the SQI (Equations (2)–(5)), were performed using Microsoft Excel. All site conditions were analyzed in a completely randomized design that considered both the site conditions and soil depths. Statistical analyses were carried out using one-way ANOVA's; and when a source of variation showed a significant effect ($p \leq 0.05$), a means separation by Tukey's was performed in order to establish differences among the means of every $S_{IND}$. Additionally, a global Pearson's correlation was conducted in order to identify possible associations among the various $S_{IND}$ ($r \geq \pm0.7$). The data were analyzed using SPSS (statistical software V11.0, Inc, Chicago, IL, USA).

## 3. Results and Discussion

### 3.1. Soil Chemical Indicators

The evaluation of the means of the chemical $S_{IND}$ showed that for each condition and soil depths there were adequate levels for pH, K, S, and Na [34] (Table 2, Table 3 and Appendix A).

**Table 2.** Soil chemical characterization results.

| Cond/Depths | pH (H$_2$O) | SOC (%) | N (%) | C/N | P * | K * | Ca ** | Mg ** |
|---|---|---|---|---|---|---|---|---|
| O$_p$ 0–5 | 5.63Aa | 14.17Aa | 0.61Aa | 23.05Aa | 2.98Aa | 109.9Aa | 1.6Aa | 0.28Aa |
| O$_p$ 5–20 | 5.59Ab | 13.33Aa | 0.61Aa | 22.09Ab | 2.08Ab | 73.8Ab | 0.46Ab | 0.16Aa |
| SO$_p$ 0–5 | 5.93Ba | 12.57Ba | 0.48Ba | 26.31Ba | 3.66Ba | 87.6Ba | 4.58Ba | 0.58Ba |
| SO$_p$ 5–20 | 5.59Bb | 11.49Ba | 0.45Ba | 25.37Bb | 2.13Bb | 62.2Ba | 1.37Bb | 0.25Ba |
| SC 0–5 | 5.9Ca | 10.82Ca | 0.46Ba | 22.37Aa | 3.63Ba | 113.7Aa | 3.23Ba | 0.32Ba |
| SC 5–20 | 5.85Cb | 13.87Cb | 0.53Ba | 21.78Ab | 2.15Bb | 70.1Ab | 3.33Ba | 0.47Ba |

* Cond, conditions; *n*:18; *p* < 0.05; * mg kg$^{-1}$; ** cmol (+) kg$^{-1}$. Distinct capital letters mean significant differences among conditions whereas lowercase letters refer to significant differences between depths.

**Table 3.** Soil chemical characterization results B. Continuation.

| Cond/Depths | Na ** | Al$_{EXCH}$ ** | ECEC | % Al$_{SAT}$ | S * | NO$_3^-$ * | NH$_4^+$ * |
|---|---|---|---|---|---|---|---|
| O$_p$ 0–5 | 0.1Aa | 0.34Aa | 2.70Aa | 12.74Aa | 8.37Aa | 16.78Aa | 10.59Aa |
| O$_p$ 5–20 | 0.1Ab | 0.26Ab | 1.16Ab | 22.09Ab | 9.23Ab | 10.27Ab | 9.07Aa |
| SO$_p$ 0–5 | 0.05Ba | 0.11Ba | 5.55Ba | 2.02Ba | 9.26Ba | 23.72Ba | 18.92Ba |
| SO$_p$ 5–20 | 0.07Bb | 0.30Bb | 2.15Bb | 13.75Bb | 10.73Bb | 15.63Bb | 13.43Ba |
| SC 0–5 | 0.08Aa | 0.14Ca | 4.06Ba | 3.54Ca | 13.20Ca | 9.16Aa | 12.85Ba |
| SC 5–20 | 0.13Ab | 0.14Ab | 4.25Bb | 3.25Cb | 11.93Cb | 12.75Ab | 15.68Ba |

* Cond, conditions; *n*:18; *p* < 0.05; * mg kg$^{-1}$; ** cmol (+) kg$^{-1}$. Distinct capital letters mean significant differences among conditions whilst lowercase letters refer significant differences between depths. *n*:18; *p* < 0.05; * mg kg$^{-1}$; ** cmol (+) kg$^{-1}$. Distinct capital letters mean significant differences among conditions whilst lowercase letters refer significant differences between depths.

The mean values of soil pH were 5.6, 5.7, and 5.9 (±0.03) (O$_p$ < SO$_p$ < SC), with significant differences measured among the SPS conditions and soil depths. Lower values of pH were observed in the 0–5 cm horizon, which could be attributed to a higher acidic condition that was favored by a greater content of OM [36], however, in all cases the pH values were within desirable ranges for supporting plant grow [34].

Soil organic C% content values at the 0–20 cm depths were 13.5, 13.1, and 11.8% (±0.28) for O$_p$, SC, and SO$_p$, respectively (Table 4). The greatest SOC% occurred in the O$_P$, despite having the lowest tree cover, as well as being the more anthropogenically affected area. This contradiction could be related to the extensive history of agricultural burns for the potato crops (*Solanum tuberosum*), thereby generating pyrogenic C, as identified by the presence of charcoal fragments and intense black color in soil samples. Pyrogenic C, is highly resistant to oxidation due to its poly aromatic structure, and therefore could be a persistent fraction of total C. In forests of *Araucaria-Nothofagus spp*, of the Tolhuaca National Park, Chile (36°52′ S y 71°56′14″ O), [37] it has been estimated that pyrogenic C represented up to 5% of the total SOC. Concerning SOC storage (0–20 cm) in the present study, stocks of 150.5, 149.8, and 143.5 Mg ha$^{-1}$ were estimated for SO$_p$, SC, and O$_p$, respectively (Figure 1).

**Table 4.** Variation of soil organic C (SOC) and total N in the period 2015–2018.

| Conditions | * SOC$_{2015}$ | SOC$_{2018}$ | * N$_{2015}$ | N$_{2018}$ | * C:N$_{2015}$ | C:N$_{2018}$ |
|---|---|---|---|---|---|---|
| O$_p$ 0–20 | 7.1 | 13.5 | 0.32 | 0.61 | 22.3 | 22.3 |
| SO$_p$ 0–20 | 7.1 | 11.8 | 0.38 | 0.46 | 19.2 | 25.6 |
| SC 0–20 | 8.0. | 13.1 | 0.42 | 0.52 | 19.5 | 22.0 |

SOC and N (%) * from Alfaro et al. (2018), weighted values * from [38].

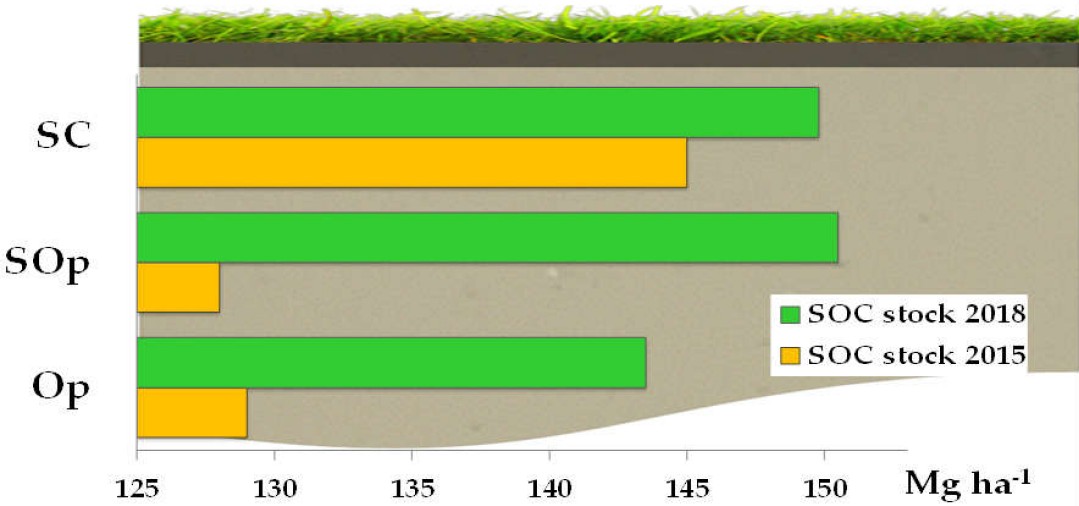

**Figure 1.** Temporal variations (2015–2018) of SOC stocks (0–20 cm) in the SPS-RA. $O_P$ + 14.5 Mg C ha$^{-1}$; $S_{OP}$ + 22.5 Mg C ha$^{-1}$; and 1.6 Mg of SOC for semi open (SO$_P$), open (O$_P$), and semi closed (SC), respectively.

The estimated carbon concentrations in the SPS-RA are significantly higher than those found in other soil conservation managements (e.g., conservation and no tillage cropping systems). In agriculturally managed Andisols, with wheat stubble incorporation from Yungay, Chile, Panichini (personal communication, May 2019), estimates indicate a mean SOC of 6.9% and a C stock of 130 Mg ha$^{-1}$. However, Muñoz et al. (2012) [39] determined SOC stocks that ranged from 33.1 to 35.5 Mg ha$^{-1}$ C after 16 years of no-tillage in volcanic soils in south-central Chile.

In our study, the total N% (0–20 cm) ranged between 0.6 and 0.5 (±0.09), where $O_p > SO_p = SC$, and were within acceptable levels. However, the bioavailable N forms had averaged values for $NO^{3-}$ that ranged from 17.7 to 11.9 mg kg$^{-1}$ (SO$_p$ > O$_p$ = SC), whereas those for $NH^{4+}$ were 15.0, 14.8, and 9.45 mg kg$^{-1}$ (SC > SO$_p$ > O$_p$, respectively), which were lower than the minimal requirements for most plants [34]. The N could be undergoing a net immobilization process in soil, or the low levels of $NH^{4+}$ could be related to the high N demand by the herbaceous component, which preferentially uptakes this particular species because of its lower energetic cost to the plant as compared with $NO^{3-}$. Alternatively, the woody component favors $NO^{3-}$ absorption because of the high soil exploratory capacity by roots as occurs in the SC condition [40,41]. Similarly, Alfaro et al. (2018) [38] found a nitrification rate pattern ($NO^{3-} \rightarrow NO_2^{-}$) of SC > SO$_p$ > O$_p$ in the SPS-RA, which was 45% higher for SC as compared with SO$_p$.

Nonetheless, it is expected that the $NH^{4+}$ levels will increase in soils, to eventually become the dominant bioavailable N species, due to the ongoing fecal depositions from the animal component within the system [42]. The temporal variation of SOC and total N are summarized in Table 4. The C/N ratio varied significantly at both ranges of depths (0–5 and 5–20 cm) as follows: O$_p$ (23.1/22.1), SO$_p$ (26.3/25.4), and SC (20.9/20.8). Ratios over 10 indicate a net immobilization of N, resulting in its incorporation into microbial biomass or by-products of microbial activity during the SOM cycling processes, consequently limiting its availability for plant growth [43].

In a previous investigation [19], the authors found a C/N ratio of 18.0 in the SPS-RA during 2014, showing a progressive increase over time of this $S_{IND}$, probably highlighting the bio constructive effects of SPS in soils. Phosphorous concentrations were 2.3–2.5 mg kg$^{-1}$ (±0.07) and corresponded to typical values in volcanic soils 3–165 mg kg$^{-1}$ [44], with a mean of 4.6 mg kg$^{-1}$ *p* for native forests [45].

However, the *p* values found in the SPS-RA were significantly lower than previously observed [46] and ranged from 15.1 to 19.9 mg kg$^{-1}$ P in volcanic soils under crop rotation of wheat (*Triticum aestivum* L.), oats (*Avena sativa* L.), subterranean clover (*Trifolium subterraneum* L.), canola (*Brassica napus* L.), and lentil (*Lens culinaris* L.) or natural grassland (6 mg kg$^{-1}$ P).

Critical P levels in the SPS-RA could be related to P fixation mediated by P-OM associations and P-Al complexes in soil [47]. For other major nutrients, K concentrations at the 0–20 cm depths were $O_p > SC > SO_p$ (82.8, 81.4, and 68.6 mg kg$^{-1}$, respectively) (±1.86), which was slightly limiting for plant growth; the S determinations showed that $SC > SO_p > O_p$ (12.2, 10.4, and 9.0 mg kg$^{-1}$, respectively) (±0.27), and were adequate soil concentrations; there were moderate to low levels of Mg $SC > SO_p > O_p$ with values of 50.5, 48.1, and 26.8 mg kg$^{-1}$ (±5.53), respectively, similar to Ca where $SC > SO_p > O_p$ (657.3, 596.2, and 215.5 mg kg$^{-1}$, respectively) (±63.1) were observed according to [34]. The measurement of soil ECEC at the 5–20 cm depths showed that $SC > SO_p > O_p$ (4.2, 3.0, and 1.5 cmol kg$^{-1}$, respectively), which was an intermediate level (except for $O_P$ 5–20 cm) with higher values in $O_P$ and $SO_P$ (0–5 cm) [48]. Regarding $Al_{EXCH}$, critical values were found of 0.3 and 0.1 cmol kg$^{-1}$ (±0.02) ($O_p = SO_p > SC$), and the $\%Al_{SAT}$ with 19.8, 10.8, and 3.3 (±1.2) for $O_p$, $SO_p$, and SC, respectively. Both $S_{IND}$ are typical for volcanic soils, but these still could lead to increased soil acidification, which would inhibit soil nutrient uptake (e.g., Ca and Mg) and restrict possible crop rotations that use Al sensitive species (e.g., barley and wheat) [49].

### 3.2. Soil Physical Indicators

In the physical $S_{IND}$ analyses, the BD, PD, $P_{OR}$, and $P_{EN}R$ were estimated to be at optimal levels in all site conditions (Table 5 and Appendix A). Individually, BD showed representative values for Andisols with 0.50, 0.53, and 0.68 g cm$^{-3}$ (±0.02) for SC, $O_p$, and $SO_p$ respectively. The lowest mean was in $O_p$ 5–20 cm (0.51 g cm$^{-3}$) and was possibly due to the presence of pyrogenic C [50], whereas higher means were observed in $SO_p$ at the 5–20 cm depths (0.65 g cm$^{-3}$). Likewise, Panichini (personal communication, May 2019) determined a mean value of BD of 0.94 g cm$^{-3}$ in volcanic soils of Yungay under stubble burning management. Moreover, PD varied from 1.95 to 2.1 g cm$^{-3}$ (±0.05) with a mean value of 2.0 g cm$^{-3}$, which was within representative ranges of volcanic soils that are rich in OM, according to Nissen et al. (2005) [51].

The same authors determined similar PD values (1.92 g cm$^{-3}$) in forest soils (0–15 cm) of the Region of Osorno, Chile. Although there were no significant differences among %WSA (50.6–49.7 and 49.5%) (±0.65), there was a tendency for $SC > O_p > SO_p$, that could be related to previous site degradation. Gradual increases of %WSA are expected to occur though, and mediated by, emerging roots and hyphae associations [52,53]. The soil compaction test (measured through its $P_{EN}R$) revealed ranges of 100–200 psi that were suitable for root anchoring/exploration and plant growth [54,55]. It should be noted that there were scattered points within $O_p$ with $P_{EN}R > 300$ psi that had a visibly reduced coverage of vegetation.

**Table 5.** Soil physical results.

| Condition/Depths | BD (g cm$^{-3}$) | PD (g cm$^{-3}$) | * $P_{OR}$ Total (%) | WHC (%) | $I_{NF}Vk$ (cm day$^{-1}$) | WSA (%) | $P_{EN}R$ (psi) |
|---|---|---|---|---|---|---|---|
| $O_p$ 0–5 | 0.6Aa | 1.9Aa | 71.2Aa | 60.8Aa | * | 49.6Aa | 100–200Aa |
| $O_p$ 5–20 | 0.5Aa | 2.0Aa | 74.2Aa | 58.9Aa | 17Aa | 49.7Aa | 100–200Aa |
| $SO_p$ 0–5 | 0.6Aa | 1.9Aa | 68.9Ba | 59.2Aa | * | 49.4Aa | 100–200Aa |
| $SO_p$ 5–20 | 0.7Aa | 2.0Aa | 68.3Ba | 52.2Aa | 18.3Aa | 49.5Aa | 100–200Aa |
| SC 0–5 | 0.5Aa | 1.9Aa | 73.6Aa | 54.4Aa | * | 51.2Aa | 100–200Aa |
| SC 5–20 | 0.5Aa | 2.0Aa | 73.9Aa | 62.9Aa | 16.2Aa | 50.4Aa | 100–200Aa |

*n*:18 and *p* < 0.05. Distinct capital letters mean significant differences among conditions, whereas lowercase letters refer significant differences between depths. * $I_{NF}V$ values were considered for the total depth (0–20 cm) and * from the Equation (1). BD, bulk density, PD, particle density, $P_{OR}$ Total, Total porosity, WHC water holding capacity, $I_{NF}Vk$, Infiltration velocity, WSA, % of water stable aggregates, $P_{EN}R$, Penetration resistance

In terms of hydraulic properties (0–20 cm), the $P_{OR}$ presented optimal values in all conditions, with SC (73.8), $O_p$ (73.5), and $SO_p$ (68.5) that promote both water storage and root development. These results are consistent with [51], who found a $P_{OR}$ of 73.9. The WHC (0–5 and 5–20 cm) was within a range of acceptable values that promote water storage and redistribution processes according to [35], and ranged as follows: $O_p$ (60.8–58.9), $SO_p$ (70.8–66), and SC (70.8–62.8) (±2.96); with the observed

gradient probably corresponding directly with the previous disturbances of the sites, and additions of SOM (in SC and $SO_P$). In the case of soil with a high $P_{OR}$ and low values of WHC ($O_p$), this could be due to hydrophobic conditions influenced by the possible presence of pyrogenic C.

The $S_{IND}$ that relates to water infiltration and percolation in soil ($I_{NF}V$) 0–20 cm, showed an intermediate level with k values of 17, 18.3, and 16.2 cm day$^{-1}$ with $SO_p > O_p > SC$ [56].

However, for $O_P$, many attempts were necessary to carry out the measurements because of compacted surface soil and hydrophobicity which impeded the proper functioning of the infiltrometer.

### 3.3. Determination of SQI

After calculating the sub-indexes for all $S_{IND}$, distinctive trends were observed in both the chemical and physical indicators (Figure 2) which illustrated some of the native (or inherent) characteristics of soil quality in volcanic soils (e.g., P, BD, $P_{OR}$, $Al_{EXCH}$, and $Al_{SAT}$). Additionally, the results showed the possible direct effects of silvopastoralism over some of the $S_{IND}$ sub-indexes (e.g., SOC). Chemical SQI was higher at $SO_P$ at the 0–5 cm depths (probably reflecting the more favorable tree density that favors the beneficial SPS interactions) and SC at the 5–20 cm depths at the less disturbed site (Table 6). Nevertheless, SOC as the most important $S_{IND}$, showed significant variations among all conditions, as evidenced in the $O_P$ condition with a higher %C content (13.5) than the $SO_P$ condition (11.8% SOC).

**Table 6.** Partial, global, and % soil quality index (SQI) scoring.

| Condition/Depths | * CHEMICAL SQI | ** PHYSICAL SQI (B) | *** GLOBAL SQI | **** SQI % |
|---|---|---|---|---|
| $O_p$ 0–5 | 13 | 64.2 | 38.6 | 22.7 |
| $O_p$ 5–20 | 7.2 | 46.4 | 26.8 | 40.9 |
| $SO_p$ 0–5 | 24.6 | 46.4 | 35.5 | 22.7 |
| $SO_p$ 5–20 | 10.1 | 42.9 | 26.5 | 27.2 |
| SC 0–5 | 10.1 | 50.0 | 30.1 | 22.7 |
| SC 5–20 | 15.9 | 64.3 | 40.1 | 22.7 |

From the Equations * (2), ** (3), *** (4), and **** (5).

The Physical SQI showed less variation among the site conditions, with the highest scores in $O_p$ at the 0–5 cm depths and SC at the 5–20 cm depths. The Global (physical plus chemical) SQI revealed less variation with scores of 26.5–40.1 from $SO_P$ at the 5–20 cm depths and $O_P$ at the 5–20 cm depths with the lowest scores in SC at the 5–20 cm depths and $O_P$ at the 5–20 cm depths. The % "critical" SQI showed that the $O_P$ at the 5–20 cm depths was the condition with the most $S_{IND}$ that were at critical levels, likely as a result of the adverse impacts of logging, over grazing, over browsing among trees, cropping, and burning practices. In addition, historic logging and fire events, erosion, and percolation losses of nutrients (e.g., P, $Ca^{2+}$, and ECEC) via leaching have likely been the causes of the low fertility levels in $O_P$ at the 5–20 cm depths [57,58].

An overall correlation analysis among the $S_{IND}$ (Table 7) revealed some possible associations as follows: the %WSA was correlated with $P_{OR}$ and $I_{NF}V$ ($r \geq 0.8$) indicating the importance of soil aggregation on hydraulic conductivity; $Al_{SAT}$ % and $Al_{EXCH}$ were correlated with pH, ECEC, $Ca^{2+}$, and $Mg^{2+}$ ($r = -0.9$) showing the inverse relationship between soil nutrients and Al forms; and SOC was correlated with N ($r = 0.9$), P ($r = 0.7$), K ($r = 0.8$), BD ($r = 0.8$), and $P_{OR}$ ($r = 0.7$) which underlined the key role of SOM in soil quality.

Thus, this study demonstrated the importance of soil quality assessment in AFS, particularly SPS, and that the calculation of simple additive linear SQI is one of the most effective methods for detecting the impacts of management practices in soils [59]. In the future, it is expected that there will be an improvement in some of the $S_{IND}$, thereby increasing SQI in the SPS-RA in the medium to long term (10–20 y), probably by the continuous thickening of the organic duff on the soil surface ($O_f$ horizon), and the widespread depositions of animal excrements, in addition to the positive effects of root and hyphae that promote the formation of stable aggregates >2.00 mm through root biomass and colonization mechanisms, which in turn creates greater SOC stabilization (e.g., C→SOC) [52,53].

## Soil quality scores for individual S$_{IND}$

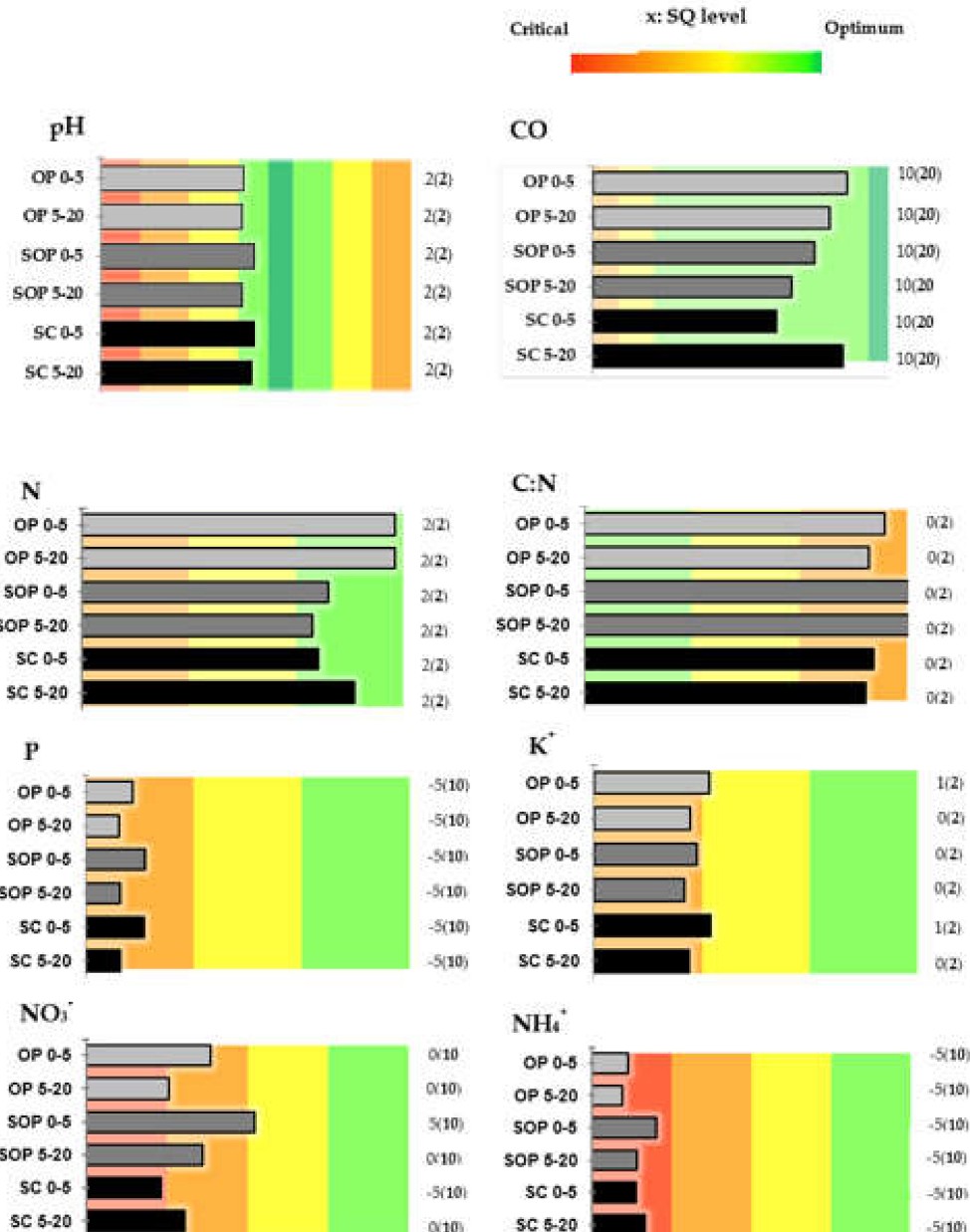

**Figure 2.** *Cont*.

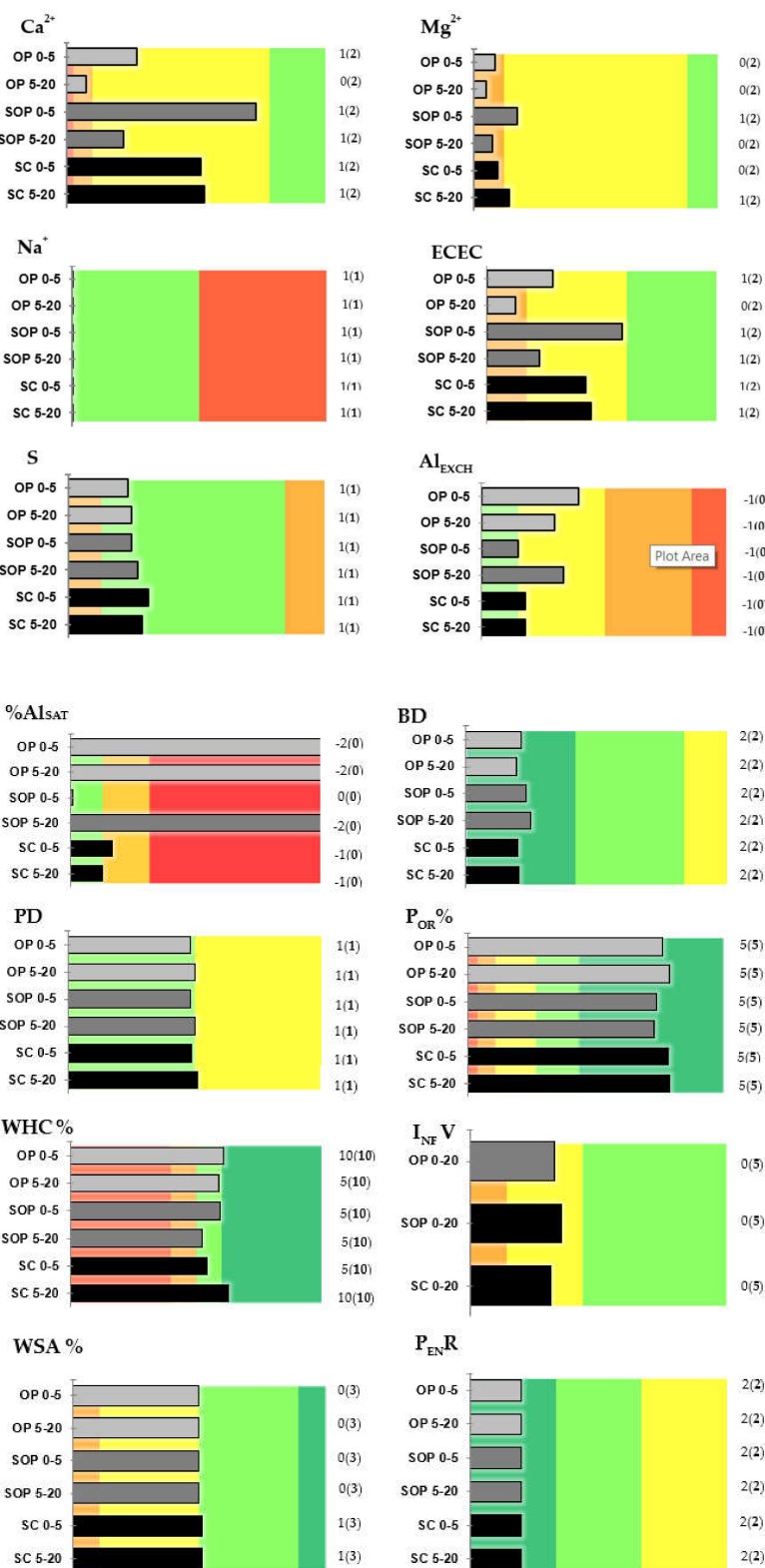

**Figure 2.** Integration of the different sub-indexes for all soil indicators (S$_{IND}$) considered. Each color corresponds to a different quality level, according to the ranges defined by Amacher et al. (2007) [34]. Greenish hues represent indexes from acceptable to optimal (differentiated from lower to greater color intensity respectively), reddish color symbolizes undesirable to critical levels; yellowish color indicates medium values of quality. Values in the columns on the right [x(**x**)], express x as the quality sub-index score, and (**x**) as the maximum possible sub-index value.

**Table 7.** Pearson's correlation coefficients for the distinct $S_{IND}$ evaluated.

| | pH | SOC (%) | N (%) | C:N | P | K | Ca²⁺ | Mg²⁺ | Na | Al_EXCH | ECEC | Al_SAT % | S | NO₃⁻ | NH₄⁺ | BD | PD | P_OR | WHC% | I_NF V | WSA% |
|---|---|---|---|---|---|---|---|---|---|---|---|---|---|---|---|---|---|---|---|---|---|
| pH | 1 | NS | NS | 0.91 | NS | NS | NS | NS | 0.8 | NS | NS | NS | NS | 0.7 | NS | 0.8 | 0.7 | NS | 0.7 | 0.8 | NS |
| SOC (%) | NS | 1 | 0 | 0.91 | 0.7 | 0.9 | NS | 0.6 | 0.8 | NS | NS | NS | NS | NS | NS | 0.8 | NS | 0.7 | NS | NS | NS |
| N (%) | NS | 0.9 | 1 | N | NS | 0.8 | NS | NS | N | NS | NS | NS | NS | 0.8 | NS | NS | NS | NS | NS | 0.6 | 0.7 |
| C:N | NS | NS | NS | 1 | NS | NS | NS | NS | N | 0.9 | NS | NS | NS | NS | NS | NS | NS | NS | 0.6 | NS | NS |
| P | NS | NS | NS | N | 1 | NS | NS | NS | N | NS | NS | NS | 0.7 | NS | NS | NS | NS | 0.6 | 0.9 | NS | 0.7 |
| K | NS | NS | NS | N | 0.8 | 1 | NS | 0.9 | 0.6 | NS | NS | NS | 0.8 | 0.9 | NS | NS | NS | 0.6 | 0.9 | NS | NS |
| Ca²⁺ | 0.9 | NS | NS | N | 0.7 | NS | 1 | NS | N | NS | NS | NS | NS | NS | NS | 0.8 | 0.8 | 0.6 | 0.7 | 0.8 | 0.6 |
| Mg²⁺ | 0.8 | NS | NS | N | NS | NS | 0.9 | 1 | N | NS | NS | NS | 0.6 | NS | NS | NS | 0.8 | NS | NS | 0.8 | 0.9 |
| Na | NS | NS | NS | −0.8 | NS | NS | NS | NS | 1 | 0.8 | NS | NS | NS | NS | NS | NS | NS | NS | NS | NS | 0.6 |
| Al_EXCH | −1 | NS | NS | N | NS | NS | −0.9 | −0.8 | N | 1 | NS | NS | NS | 0.7 | NS | 0.8 | 0.7 | NS | 0.7 | 0.8 | NS |
| ECEC | 0.9 | NS | NS | N | 0.7 | NS | 1 | 0.9 | N | −0.9 | 1 | NS | NS | NS | NS | 0.8 | 0.9 | 0.6 | 0.6 | 0.8 | 0.7 |
| Al_SAT % | −0.9 | NS | NS | N | −0.6 | NS | −0.9 | −0.9 | N | 0.9 | −0.9 | 1 | NS | NS | NS | 0.9 | 0.7 | 0.8 | 0.8 | 0.7 | NS |
| S | 0.6 | −0.6 | NS | N | NS | NS | NS | NS | N | −0.6 | NS | NS | 1 | NS | NS | 0.6 | NS | NS | NS | NS | NS |
| NO₃⁻ | NS | NS | NS | 0.7 | NS | NS | NS | 0.6 | N | NS | NS | NS | NS | 1 | NS | NS | 0.7 | NS | 0.7 | NS | NS |
| NH₄⁺ | 0.8 | NS | NS | N | NS | NS | 0.9 | 0.9 | N | −0.8 | 0.9 | −0.8 | NS | 0.7 | 1 | NS | 0.7 | NS | 0.8 | 0.9 | 0.9 |
| BD | NS | NS | NS | N | NS | NS | NS | NS | N | NS | NS | NS | NS | NS | NS | 1 | 0.6 | 0 | NS | NS | 0.7 |
| PD | NS | NS | NS | N | NS | NS | NS | NS | N | NS | NS | NS | NS | NS | NS | NS | 1 | NS | 0.6 | 0.6 | NS |
| P_OR | NS | NS | NS | N | NS | NS | NS | NS | N | NS | NS | NS | NS | −0.6 | NS | −0.7 | NS | 1 | NS | 0.7 | 0.9 |
| WHC% | NS | 0.6 | 0.6 | N | NS | NS | NS | NS | N | NS | NS | NS | NS | N | NS | NS | NS | NS | 1 | NS | 0.9 |
| I_NF V | NS | NS | NS | N | NS | NS | NS | NS | N | NS | NS | NS | NS | N | NS | NS | NS | NS | NS | 1 | NS |
| WSA% | NS | NS | NS | N | NS | NS | NS | NS | N | NS | NS | NS | NS | N | NS | NS | NS | NS | NS | NS | 1 |

Among remarkable correlations are the following: pH-ECEC ($r = 0.9$), pH and the cations Ca²⁺ and Mg²⁺ ($r = 0.9, 0.8$), pH-Al_EXCH ($r = −1$), pH-Al_SAT% ($r = −0.9$), these last inverse correlations demonstrating the importance of Al in nutrient availability. Complementary correlations such as Al_EXCH, Ca²⁺ ($r = −0.9$), Mg²⁺ ($r = −0.8$), S ($r = 0.7$), as with Al_SAT% and ECEC, Ca²⁺, and Mg²⁺ ($r = −0.9$) showed the dominance of this element in the interchange sites as long as pH values decrease, thus substituting these cations. SOC correlated with different nutrients, i.e., N ($r = 0.9$), P ($r = 0.7$), K ($r = 0.8$) as well as the physical $S_{IND}$ BD ($r = 0.8$) and P_OR ($r = 0.7$), which highlighted its crucial role on nutrient supply and particle arrangement.

## 4. Conclusions

An overall chemical SQI was calculated that showed a large variability over all site conditions, with $NH^{4+}$, $NO^{3-}$, and P, being the most limiting chemical indicators ($S_{IND}$). Regarding the overall physical SQI, there were no significant differences between individual physical $S_{IND}$ (except for $P_{OR}$ in $SO_p$), presumably due to the timescale by which those properties underwent changes. The combined (chemical + physical) Global SQI (0–20 cm), showed the SC condition had the highest SQI (37.6), followed by $O_p$ (29.8) and $SO_P$ (28.8), demonstrating the importance of trees in preserving soil quality. However, the highest SQI was in SC at the 5–20 cm depths, reflecting its history of less disturbed management. There was an estimated increase of SOC stock of about 7.5, 4.8, and 1.6 Mg ha$^{-1}$yr$^{-1}$, in addition to the total N increase of 0.5, 2.0, and 1.2 kg ha$^{-1}$ for $SO_p$, $O_p$, and SC, respectively. Nonetheless, $NO_3^-$ and $NH_4^+$ availability is limited, which is strongly linked to the progressive increase of C/N ratios that was observed. These preliminary results confirm the importance of the SPS-RA in the C→SOC sequestration process, with results that were generally within the typically reported values in the literature (only occasionally exceeding them, e.g., $SO_P$) and showing that these systems promote nutrient cycling and soil restoration. Future seasonal and long-term research is now required in order to understand the role of biological activity in SOM transformations and determine the soil C balances in order to elucidate the possible C stabilization processes involved.

**Author Contributions:** Conceptualization, F.D. and J.O.; methodology, M.P., J.O., P.N., and F.D.; validation, F.D., J.O., E.Z., and N.B.S.; investigation, J.O. and P.N.; resources, M.P. and J.O.; formal analysis and data curation, J.O.·and M.P.; writing—original draft preparation, J.O. and P.N.; writing—review and editing, F.D., J.O., and N.B.S.; visualization, J.O.; supervision, F.D., N.B.S., M.P., E.Z., and P.A.M.-H.; project administration, F.D., J.O., and M.P.; funding acquisition, F.D. and M.P. All authors have read and agreed to the published version of the manuscript.

**Funding:** This study was funded by the Native Forest Research Fund of the National Forestry Corporation, Chile (FIBN-CONAF project No. 001–2014), VRID-UDEC Multidisciplinary project No. 219.142.040-M, Chile and the INIA Quilamapu agroecology lab.

**Acknowledgments:** We wish to express our sincere thanks to the Ranchillo Alto Research & Teaching Forest of the University of Concepción, and to INIA Quilamapu for full access to their agroecology lab.

**Conflicts of Interest:** The authors declare no conflict of interest.

## Appendix A

Sub-index values for each $S_{IND}$ considered.

Below are shown $S_{IND}$ ranges or levels of soil quality and the correspondent sub-index values (modified from [34]).

**Table A1.** Soil quality levels and their associated sub-index values for physical $S_{IND}$.

| Physical $S_{IND}$ | Level | Interpretation | Subindex | Source |
|---|---|---|---|---|
| PD (g cm$^{-3}$) | <2 | Desirable | 1 | [51] |
| | >2 | Without effect | 0 | |
| BD (g cm$^{-3}$) | <1.10 | Optimum | 2 | [60] |
| | 1.10–1.47 | Desirable | 1 | |
| | >1.47 | Low | 0 | |
| $P_{OR}$ (%) | <5 | Critical | −5 | [61] |
| | 5–10 | Restrictive | 0 | |
| | 10–25 | Acceptable | 1 | |
| | 25–40 | Desirable | 2 | |
| | >40 | Optimum | 5 | |

**Table A1.** *Cont.*

| Physical $S_{IND}$ | Level | Interpretation | Subindex | Source |
|---|---|---|---|---|
| WHC (%) | >60 | Optimum | 10 | [35] |
| | 51–60 | Acceptable | 5 | |
| | 41–50 | Low | 0 | |
| | <40 | Critical | −10 | |
| $I_{NF}Vk$ (cm day$^{-1}$) | <8.64 | Undesirable | −5 | [56] |
| | 8.64–20 | Acceptable | 0 | |
| | 20–43.2 | Optimum | 5 | |
| $P_{EN}R$ (psi) | >300 | Undesirable | 0 | [54,55] |
| | 200–300 | Acceptable | 1 | |
| | 100–200 | Optimum | 2 | |
| WSA (%) | <50 | Undesirable | 0 | [54] |
| | 50–70 | Medium | 1 | |
| | 70–90 | High | 2 | |
| | >90 | Optimum | 3 | |

**Table A2.** Soil quality levels and their associated sub-index values for chemical $S_{IND}$.

| Chemical $S_{IND}$ | Level | Interpretation | Sub-index | Source |
|---|---|---|---|---|
| pH | <3.0 | Super critical | −1 | [34] |
| | 3.01–4.0 | Critical | 0 | |
| | 4.01–5.5 | Limiting | 1 | |
| | 5.51–6.8 | Desirable | 2 | |
| | 6.81–7.2 | Optimum | 2 | |
| | 7.21–7.5 | Acceptable | 1 | |
| | 7.51–8.5 | Limiting | 1 | |
| | >8.5 | Critical | 0 | |
| SOC (%) | >15 | Excellent | 20 | [35] |
| | 5–15 | High | 10 | |
| | 3–5 | Moderate | 1 | |
| | <2 | Low | −10 | |
| N (%) | >0.5 | Desirable | 2 | [34] |
| | 0.1–0.5 | Adequate | 1 | |
| | <0.1 | Insufficient | 0 | |
| $NO_3^-$ (mg kg$^{-1}$) | <10 | Critical | −5 | [35] |
| | 10–20.1 | Insufficient | 0 | |
| | 20.1–40 | Adequate | 5 | |
| | >40 | Desirable | 10 | |
| $NH_4^+$ (mg kg$^{-1}$) | <25 | Critical | −5 | [35] |
| | 25–50 | Insufficient | 0 | |
| | 51–75 | Adequate | 5 | |
| | >75 | Desirable | 10 | |
| | 20–20 | Moderate | 1 | |
| | >20 | Insufficient | 0 | |

**Table A2.** *Cont.*

| Chemical $S_{IND}$ | Level | Interpretation | Sub-index | Source |
|---|---|---|---|---|
| C:N ratio | 1–10 | Adequate | 2 | [34] |
| | 10–20 | Moderate | 1 | |
| | >20 | Insufficient | 0 | |
| P (mg kg$^{-1}$) | >16 | Adequate | 10 | [48] |
| | 5–15 | Moderate | 1 | |
| | <5 | Insufficient | −5 | |
| K (mg kg$^{-1}$) | >500 | Adequate | 2 | [34] |
| | 100–500 | Moderate | 1 | |
| | <100 | Insufficient | 0 | |
| S (mg kg$^{-1}$) | >100 | Insufficient | 0 | [34] |
| | 1–100 | Adequate | 1 | |
| | <1 | Insufficient | 0 | |
| Ca (mg kg$^{-1}$) | >1000 | Desirable | 2 | [34] |
| | 101–1000 | Adequate | 1 | |
| | 10–100 | Insufficient | 0 | |
| | <10 | Critical | −1 | |
| Mg (mg kg$^{-1}$) | >500 | Adequate | 2 | [34] |
| | 50–500 | Moderate | 1 | |
| | <50 | Insufficient | 0 | |
| ECEC(cmol kg$^{-1}$) | >6.27 | Adequate | 2 | [62] |
| | 1.65–6.27 | Moderate | 1 | |
| | <1.65 | Insufficient | 0 | |
| Exchangeable % Na | <15 | Critical | 0 | [34] |
| | ≤15 | Acceptable | 1 | |
| Al$_{EXCH}$(cmol kg$^{-1}$) | <0.1 | Adequate | 0 | [62] |
| | 0.11–0.51 | Moderate | −1 | |
| | 0.51–0.81 | Undesirable | −2 | |
| | >0.81 | Critical | −3 | |
| Sat Al (%) | 1.1–3.1 | Adequate | 0 | [62] |
| | 3.2–6.1 | Moderate | −1 | |
| | 6.2–12 | High | −2 | |
| | >12 | Critical | −5 | |

## Appendix B

**Table A3.** List of abbreviations.

| Abbreviation | Description | Abbreviation | Description |
|---|---|---|---|
| Gg | Gigagrams | NO$_3^-$ | Nitrate |
| Pg | Petagrams | P | Phosphorous |
| AFS | Agroforestry system | K$^+$ | Potassium |

**Table A3.** *Cont.*

| Abbreviation | Description | Abbreviation | Description |
|---|---|---|---|
| SPS | Silvopastoral systems | $Ca^{2+}$ | Calcium |
| SPS-RA | Silvopastoral systems Ranchillo Alto | $Mg^{2+}$ | Magnesium |
| C | Carbon | $Na^+$ | Sodium |
| $CO_{2eq}$ | Carbon equivalent | S | Sulphur |
| SOM | Soil organic matter | $Al_{EXCH}$ | Exchangeable Al |
| SOC | Soil organic carbon | $\%Al_{SAT}$ | % of Al saturation |
| C→SOC | Carbon sequestration | ECEC | Effective cation-exchange capacity |
| SQ | Soil quality | pH | Soil reactivity |
| SQI | Soil quality index | PD | Particle density |
| $S_{IND}$ | Soil quality indicator | BD | Bulk density |
| $O_p$ | Open condition | % (POR) | Total porosity |
| $S_{Op}$ | Semi-open condition | WSA | % of water stable aggregates |
| SC | Semi-closed condition | INFV | Infiltration velocity |
| N | Nitrogen | WHC | Water holding capacity |
| C/N | Carbon-to-nitrogen ratio | $P_{EN}R$ | Penetration resistance |
| $NH_4^+$ | Ammonium | | |

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
