# Peer review of "Soil Quality Changes within a (Nothofagus obliqua) Forest Under Silvopastoral Management in the Andes Mountain Range, South Central Chile"

_sustainability, doi:10.3390/su12176815_

Round 1
Reviewer 1 Report
The manuscript however interesting suffers from some serious problems. Most importantly is that the English need a thorough revision. Together with the extensive use of abbreviations the language hampers reading and make the judging the merits of the manuscript quite difficult.
The sampling soil sampling procedure needs to be extended, for example it is not fully clear about the number of samples used in the statistical analys.
Examples Minor comments
Line 21 – remove ’only’
Line 27 – Clarify the notification
Line 30 – spell out abbreviation (Sind)
Line 37 – Change ‘greenhouse emissions’ to ‘greenhouse gas emissions’
Line 30 – change ‘for’ to ‘of’
Line 44 – I prefer the term ‘Climate change’
Line 110 – Rephrase sentence
Line 119 – change ‘proposed by’ to according to’
Reviewer 2 Report
Effects of a newly established silvopastoral system over soil quality in an old, degraded Roble (Nothofagus obliqua) forest in the Andes mountain range, south central Chile
Ortiz, J, Dube F., Neira P., Panichini M., Stolpe N.B., Zagal, E., Martinez_Hernandez, P.A.
General Comments
The authors have conducted a field study in south-central Chile. This 5-year study examined 3 sites (Op, Sop and SC) and two soil horizons (0-5 cm and 5-20 cm). The measured parameters and calculated indices showed the importance of trees for soil quality.
The suggested specific comments are meant to improve the flow of this manuscript so that the authors’ work is clearly communicated.
Specific Comments
Lines 2-5: Consider changing/simplifying the title for more impact. A suggestion follows: Change title to “Soil Quality Changes within a Degraded Roble (Nothofagus obliqua) Forest in the Andes Mountain Range, south-central Chile
Line 7: Change font size of last author (Martinez_Hernandez, P.A.) to be consistent with previous authors
Line 34: Change “recuperation” to “reclamation.”
Line 40: Delete “processes”
Line 43: Change preposition “from” to ‘in” and change “about” to “approximately”.
Line 47: Change “of crucial relevance” to “critically relevant”.
Line 47-48: Change “soil natural processes” to “natural pedologic processes.
Line 48: Change “management proposes” to “management practices”. If the suggested change does not capture your intent, then please clarify.
Line 48: Change “environmental services” to “environmental functions”. If the suggested change does not capture your intent, then please clarify.
Line 48: Insert author’s name (i.e. Karlen and others or Karlen et.al.,) along with reference [9].
Line 51: It is best to start the sentence with the words “Soil Quality”. The shorthand, SQ, can be used within a sentence i.e. “SQ indicators” instead of “soil quality indicators”.
Line 52-53: Consider changing the sentence as follows: Delete “referred as properties-processes”; Delete “in an”; Change “indirect way” to “indirectly”.
Line 55: Change “showing sensibility” to “to show seasonal or atmospheric variations and have realistic management practices”.
Line 56: Delete “feasible” and insert “to be” before “useful”.
Line 90: Insert author’s name along with reference (22).
Line 90: Change “of Andand isol order” to “Andisols”.
Line 92-97: This seems to be a run-on sentence and the message is not clear. First state the two most important concepts (agricultural production and the hydrologic cycle), then discuss.
Suggest changing sentence as follows: (1). Insert “and ii) water cycling as snow and rain.” after “i) agricultural production”; (2). Begin next sentence with “Agricultural production represents approximately 60% of national arable land. (3). Start next sentence as follows “The Andean mountain range [23] contains between 30-70% total surface area of Andisols (2.5Mha).” If these proposed changes do not capture your intent, please modify these suggestions and simplify the sentence for clarity and for the reader.
Line 97: Suggest inserting “adverse” before “impacts” and deleting “degradative”.
Line 98: By the term “canaliculus”, do you mean rill or gully erosion? If you mean rill and gully
erosion, them replace “canaliculus via water erosion” with “rills” or “gullies”. Gullies are larger than rills.
Line 101: Change “chemical parameters :” to “chemical parameters such as”.
Line 103: Change “physical parameters :” to “physical parameters such as”.
Line 111: Change “individuals” to “individual analyses.”
Line 116: Insert author’s name along with reference [27].
Line 119: Insert author’s name along with reference [28].
Line 120: Insert author’s name along with reference [29].
Line 141: Change overwritten text for readability.
Line 143: Insert author’s name along with reference [30]
Line 151: Insert author’s name along with reference [31]
Lines 168 and 170: Insert author’s name along with reference [34]
Line 225: Insert author’s name along with reference [38]
Line 228: Change “whilst” to “whereas”.
Line 229: Delete “both” before being.
Line 233: Change “whilst” to “whereas”.
Line 236: Insert “of” after “because”.
Line 238: Insert author’s name along with reference [41].
Lines 270-271: May delete “by” instead of inserting author’s name. Also suggest deleting “, that” as well as inserting “and”.
Lines 293-295: Delete “Despite”, capitalize “Both”, insert “but” after “soils,” and delete “processes” after “acidification”.
Line 301: Delete “,” and insert “.” To end sentence. Capitalize “The” to begin next sentence.
Line 317: Insert author’s name along with reference [51]. Also, delete “(previously mentioned)”.
Line 318: Delete preposition“to” before “[35]”.
Line 325-327: The authors mention a very important observation (compacted ground) during attempted measurements at open site, Op.
Line 352: Change “ot” to “at”?
Lines 358-418: Figure 2. It is not clear what the x values are on these graphs. Can the x-values be more clearly shown?
Line 433: Change “the preceding degradative processes including” to “adverse impacts such as”.
Line 434: Delete “due to”.
Line 440: Delete “antagonism or”.
Line 475: Insert the preposition “of” between “thickening” and “the”.
Round 2
Reviewer 1 Report
No further comments